# Edge-Enhanced with Feedback Attention Network for Image Super-Resolution

**DOI:** 10.3390/s21062064

**Published:** 2021-03-15

**Authors:** Chunmei Fu, Yong Yin

**Affiliations:** School of Microelectronics and Communication Engineering, Chongqing University, Chongqing 400044, China; fuchunmei@cqu.edu.cn

**Keywords:** image super-resolution, channel attention, spatial attention, feedback, edge enhance

## Abstract

Significant progress has been made in single image super-resolution (SISR) based on deep convolutional neural networks (CNNs). The attention mechanism can capture important features well, and the feedback mechanism can realize the fine-tuning of the output to the input. However, they have not been reasonably applied in the existing deep learning-based SISR methods. Additionally, the results of the existing methods still have serious artifacts and edge blurring. To address these issues, we proposed an Edge-enhanced with Feedback Attention Network for image super-resolution (EFANSR), which comprises three parts. The first part is an SR reconstruction network, which adaptively learns the features of different inputs by integrating channel attention and spatial attention blocks to achieve full utilization of the features. We also introduced feedback mechanism to feed high-level information back to the input and fine-tune the input in the dense spatial and channel attention block. The second part is the edge enhancement network, which obtains a sharp edge through adaptive edge enhancement processing on the output of the first SR network. The final part merges the outputs of the first two parts to obtain the final edge-enhanced SR image. Experimental results show that our method achieves performance comparable to the state-of-the-art methods with lower complexity.

## 1. Introduction

Single image super-resolution (SISR) is a classic computer vision task, which aims to use a low-resolution (LR) image to reconstruct the corresponding high-resolution (HR) image. Image super-resolution is an ill-posed problem since an LR image can be reconstructed to obtain multiple HR images, and the reconstructed solution space is not unique. At present, numerous image SISR methods have been proposed, which can be classified as interpolation-based methods [1], reconstruction-based methods [2,3], and learning-based methods [4,5,6,7,8].

In recent years, the convolutional neural networks (CNNs) composed of multiple convolutional layers have benefited from the number and size of the convolutional kernel of each convolutional layer, which gives them a powerful ability of expression and learning. Dong et al. [4] introduced a three-layer end-to-end convolutional neural network (SRCNN) to implement the image SR and pioneeringly realized the application of deep learning in image SR. Since then, the deep learning-based methods have attracted widespread attention because of their super reconstruction performance. However, the high computational cost limits the practical application of SRCNN. The Fast SR Convolutional Neural Networks (FSRCNN) [7] proposed later obtains better reconstruction performance with less computational cost. Research shows that the two key factors of the deep network depth and skip connections can improve SR reconstruction performance to a certain level. Therefore, to fully use the deep advantages of CNN, Accurate Image SR Using Very Deep Convolutional Networks (VDSR) [5] was proposed, which increased the depth of the convolutional layer to 20 and used skip connections and greatly improved the peak signal-to-noise ratio (PSNR) [9] and visual quality. However, the problem of gradient vanishing/explosion is serious with the increase in network depth, and it is difficult to converge in training. To address the above issues and further improve the reconstruction performance, a series of deeper networks based on residual learning have been proposed, such as that proposed in [7]. Moreover, inspired by Densely Connected Convolutional Networks (DenseNet) [10], SRDenseNet [11], Enhanced Deep Residual Networks for SISR (EDSR) [12], and Residual Dense Network for Image SR (RDN) [13] have also been successfully proposed. They directly connect each layer with subsequent layers and provide an effective method for the reuse of feature maps. Similar to most traditional deep learning-based methods, they share information in a feed-forward manner. However, the feed-forward method prevents the effective information of the latter layer from being fed back to the previous layer, and the input of the previous layer cannot be adjusted. Hence, recent studies [14,15] have applied the feedback mechanism to the network architecture. In the theory of human recognition, people can always find the correlation between them based on the original data and focus on some of its important features. Inspired by this phenomenon, many studies have applied the attention mechanism [16]. Zhang et al. [17] proposed a Very Deep Residual Channel Attention Networks (RCAN), and for the first time introduced the attention mechanism into the image SR, and obtained a high PSNR, but the network model was too complex, and the real-time performance was poor. Li et al. [18] proposed Multi-scale Residual Network for Image SR (MSRN), which uses the context information of spatial features and feature fusion technology to connect the outputs of all residual blocks, instead of increasing the depth of the CNN to improve the reconstruction performance. Although these methods have introduced an attention mechanism and have achieved certain results, they are only simple applications of the attention mechanism originally used for natural language processing (NLP). They still cannot full use multi-scale feature information, and they have not found a more suitable attention mechanism method for image SR. Inspired by the work in [19,20], we proposed a lightweight attention module for image SR to solve the above problems. Experiments show that our model has achieved better performance.

Almost all the existing methods train the model by minimizing the mean square error (MSE) or L1 loss between the reconstructed HR image and the ground truth. Even if high PSNR values are obtained, problems of over-smoothing and edge blurring are unavoidable. To obtain better image perceptual quality, super-resolution generative adversarial network (SRGAN) [21] proposed a perceptual loss function, which includes adversarial loss [22] and content loss calculated on the feature maps of the Very Deep Convolutional Networks for Large-Scale Image Recognition (VGG) network [23]. Unfortunately, the PSNR of this method is much lower than other methods. To obtain better image edges without sacrificing PSNR, we propose an edge detection and enhancement network (EdgeNet) inspired by [24,25].

In summary, our major contributions are as follows:We propose an edge enhanced feedback attention image super-resolution network (EFANSR), which comprises three stages: a dense attention super-resolution network (DASRNet), an edge detection and enhancement network (EdgeNet), and a fusion reconstruction module. The EdgeNet performs edge enhancement processing on the output image of DASRNet, and then the final SR image is obtained through the final fusion module.In DASRNet, we propose a spatial attention (SA) block to re-check the features and make the network pay more attention to high-frequency details and a channel attention (CA) block that can adaptively assign weights to different types of feature maps. We also apply a feedback mechanism in DASRNet. The feedback mechanism brings effective information of the latter layer back to the previous layer and adjusts the input of the network.We propose an EdgeNet that is more suitable for image SR. It extracts edge feature information through multiple channels and fully uses the extracted edge information to reconstruct better clarity and sharper edges.

We organize the remainder of this paper as follows: The works related to our research are presented in Section 2. The network structure and methods are described in Section 3. Section 4 discusses the performance of different loss functions and the differences in the works most relevant to our research, and the conclusions are given in Section 5.

## 2. Related Works

### 2.1. Deep Learning-Based Image Super-Resolution

Deep learning has shown powerful advantages in various fields of computer vision, including image SR. In 2014, Dong et al. [4] proposed a three-layer convolutional neural network (SRCNN), which applied deep learning to image SR for the first time. Compared with other traditional image SR methods, the reconstruction performance of SRCNN is significantly improved, but the extremely simple network structure limited its expressive ability. Inspired by the VGG [25], Kim et al. [5] increased the depth of CNN to 20 layers so that the network could extract more feature information from LR images. VDSR [5] used residual learning to ease the difficulty of deep network training and achieved considerable performance. To improve the presentation ability of the model while reducing the difficulty of network training, some recent works have proposed different variants of skip connections. The works in [6,17,18] use the residual skip connection method proposed in [12]. The works in [10,13,26,27] use the dense skip connection method proposed in [11].

Although these methods used skip connections, each layer can only receive feature information from the previous layers, which lacks enough high-level contextual information and limits the network’s reconstruction ability. In addition, the existing research treats both spatial and channel features equally, which also limits the adaptive ability of the network when processing the features. The prime information lost in the image down-sampling process is concentrated on the details such as edges and textures. However, none of the previous methods have a module that can contain as much high-frequency detail information as possible to process features. Therefore, it is very necessary to establish an attention mechanism that is more suitable for image SR tasks. Moreover, the edge blur in the image SR is still a prominent problem, and it is also extremely important to design an SR method that can improve the edge quality of the reconstructed image.

### 2.2. Feedback Mechanism

The feedback mechanism allows the network to adjust the previous input through feedback output information. In recent years, the feedback mechanism has also been used in many network applications of computer vision tasks [15,28]. For image SR, Haris et al. [29] proposed an iterative up and down projection unit based on back-projection to realize iterative error feedback. Inspired by Deep Back-Projection Networks for SR (DBPN) [29], Pan Z et al. [30] proposed Residual Dense Backprojection Networks (RDBPN) using the residual deep back-projection structure. However, these methods do not achieve a genuine sense of feedback; the information flow in the network is still feedforward. Inspired by [14], we designed a dense feature extraction module with a feedback mechanism.

### 2.3. Attention Mechanism

Attention refers to the mechanism by which the human visual system adaptively processes information according to the characteristics of the received information [31]. In recent years, to improve the performance of the model, when dealing with complex tasks, the attention mechanism has been widely applied in high-level computer vision tasks, such as image classification [32]. However, there are few applications in image SR because simply even applying the attention mechanism in low-level computer vision tasks can decrease the performance. Therefore, it is very important to establish an effective attention mechanism for image SR tasks.

### 2.4. Edge Detection and Enhancement

Image edge detection is a basic technology in the field of computer vision. How to quickly and accurately obtain image edge information has always been a research hotspot and has also been widely studied. Early methods focused on color intensity and gradient, as was done by Jones [33]. The accuracy of these methods in practical applications still needs to be further improved. Since then, methods based on feature learning have been proposed, which usually use complex learning paradigms to predict the magnitude of the edge point gradient. Although they have better results in certain scenarios, they are still limited in edge detection that represents high-level semantic information.

Recently, to further improve the accuracy of edge detection, numerous edge detection methods based on deep learning have been proposed, such as Holistically-nested edge detection (HED) [34] and Richer Convolutional Features for Edge Detection (RCF) [23]. The problem of edge blur in image SR is very prominent. Before Kim et al. proposed SREdgeNet [24], no other image SR methods used edge detection to solve this problem. SREdgeNet combines edge detection with image SR for the first time, enabling super-resolution reconstruction to obtain better edges than other super-resolution methods.

SREdgeNet uses dense residual blocks and dense skip connections to design the edge detection module, DenseEdgeNet. The network is too complex with huge network parameters, which consume a lot of storage space and training time and leads to poor real-time performance. To address the above problems, we proposed EdgeNet, which is a lightweight edge detection network comprising only three convolution paths and two pooling layers. This design greatly reduces the complexity of the network, and it can also full use the multi-scale information of the characteristic channel to generate more accurate edges.

## 3. Proposed Methods

We show the framework of our proposed EFANSR in Figure 1. EFANSR can be divided into three parts: DASRNet, EdgeNet, and the final fusion part. Let $I_{\mathrm{LR}}$ and $I_{\mathrm{SR}}$ represent the input and output images of our network, respectively.

DASRNet takes *I*_LR_ as input and up-samples it to the desired output size as the following expression
(1)$$I_{\mathrm{sr}}=\mathrm{SR}\left(I_{\mathrm{LR}}\right),$$
where SR(⋅) represents all operations performed in DASRNet. Our EdgeNet predicts the edge information of the up-sampled SR image output by DASRNet and enhances its edge. The expression is as follows
(2)$$I_{\mathrm{edge}}=E\left(I_{\mathrm{sr}}\right),$$
where $I_{\mathrm{edge}}$ is the output of EdgeNet, and E(⋅) denotes the functions of EdgeNet. The Fusion part obtains the final SR image of the entire super-resolution network by fusing $I_{\mathrm{sr}}$ and $I_{\mathrm{edge}}$
(3)$$I_{\mathrm{SR}}=F\left(I_{\mathrm{sr}},I_{\mathrm{edge}}\right),$$
where F(⋅) represents the operation of Fusion to generate the final SR image.

### 3.1. DASRNet

We show the architecture of our DASRNet in Figure 2a, which can be divided into three parts: shallow feature extraction, deep feature extraction, and up-sampling reconstruction. In this section, we use $I_{\mathrm{LR}}$ and $I_{\mathrm{sr}}$ to denote the input and output of DASRNet.

The shallow feature extraction part comprises a Conv layer and a 1 × 1 Conv layer. Herein, “Conv” and “1 × 1 Conv” both represent a convolutional layer, the number of filters is 64, stride size is 1, while the kernel sizes are 3 × 3 and 1 × 1, respectively. We use $F_{s}$ to represent the output feature maps of the Conv layer,
(4)$$F_{s}=F_{sfe1}\left(I_{\mathrm{LR}}\right)=W_{s}\times I_{\mathrm{LR}},$$
where $F_{sfe1}\left(\cdot \right)$ refers to Conv operation, $W_{s}$ denotes the filters and the biases items are omitted for simplicity. $F_{s}$ is transmitted as input to the 1 × 1 Conv layer, and the output is represented by $F_{0}$
(5)$$F_{0}=F_{sfe2}\left(F_{s}\right)=W_{0}\times I_{\mathrm{LR}},$$
where $F_{sfe2}\left(\cdot \right)$ refers to the 1 × 1 Conv operation and $F_{0}$ serves as the input of the later deep feature extraction module.

As shown in the highlighted green in Figure 2a, the deep feature extraction part contains *N* dense residual modules with spatial attention and channel attention, for which we use dense spatial and channel attention (DSCA) *i* (i=1, 2, …,N) to denote each of them.

Our work contains a total of *T* iterations, and we use *t*
(t=1, 2, …,T) to denote any one of them. During the t th iteration, the output of the (t−1) th iteration of the N th DSCA module $F_{N}^{t-1}$ is taken as one of the inputs of the t th iteration of the first DSCA module. The output of the t th iteration of the N th DSCA module is represented as $F_{N}^{t}$, which can be obtained by the following expression:(6)$$F_{N}^{t}=F_{N}\left(F_{N-1}^{t}\right)=F_{N}\left(F_{N-1}\left(\cdots \left(F_{i}\left(F_{i-1}\left(\cdots \left(F_{1}\left(F_{0},F_{N}^{t-1}\right)\right)\cdots \right)\right)\right)\cdots \right)\right),$$
where $F_{0}$ is one of the inputs of DSCA 1, $F_{i}\left(\cdot \right)$ represents a series of operations performed in the i th DSCA. We will elaborate on DSCA in Section 3.2. Inspired by [12], we adopt global feature fusion (GFF) and local feature fusion (LFF) technology to fuse the extracted depth features. The fusion output $F_{f}^{t}$ can be obtained by
(7)$$F_{f}^{t}=F_{\mathrm{conv}}\left(concat\left(F_{1}^{t},F_{2}^{t},\cdots ,F_{i}^{t},\cdots ,F_{N}^{t}\right)\right),$$
where $F_{\mathrm{conv}}\left(\cdot \right)$ represents the 1 × 1 Conv operation. $F_{f}^{t}$ is then added with the up-sampled features.
(8)$$F^{t}=F_{lrup}\left(I_{\mathrm{LR}}\right)⨁F_{f}^{t},$$
where ⨁ represents element-wise summation, and $F_{lrup}\left(\cdot \right)$ indicates the up-sampling operation. Considering the reconstruction performance and processing speed, we choose the bilinear kernel as the up-sampling kernel.

Our up-sampling reconstruction part adopts the sub-pixel convolution proposed by [35] and the generated SR image $I_{\mathrm{sr}}$ without edge enhancement can be obtained as the following expression
(9)$$I_{\mathrm{sr}}=F_{↑}\left(F^{t}\right),$$
where $F_{↑}\left(\cdot \right)$ denotes the upscale operator.

### 3.2. Dense Spatial and Channel Attention (DSCA) Block

The structure of the dense spatial and channel attention module (DSCA) we proposed is shown in Figure 2b, and the structure of SA and CA blocks are shown in Figure 3a,b, respectively.

**Spatial Attention block.** We design a new spatial attention mechanism to accurately reconstruct the detailed information of the high-frequency region. The whole calculation process is shown in Figure 3a. In contrast to other spatial attention mechanisms suitable for high-level vision tasks, our SA block consists of 3 Conv layers, 3 Deconv layers, and 2 symmetric skip connections, without pooling layers. Gradient information can be transferred directly from the bottom layer to the top layer through the skip connection, which alleviates the problem of vanishing gradient. The stacked convolutional layers allow our network to have a larger receptive field. Thus, the contextual information is fully utilized.

For a given input feature $f_{e}$, a 2D attention mask $f_{sa}$ is obtained after passing the SA block. The final output of the SA block $F_{sa}$ can be obtained by
(10)$$F_{sa}=f_{e}\times f_{sa},$$

**Channel Attention block.** We show the structure of our CA block in Figure 3b. It includes a global average pooling (GAP) layer and two “1 × 1 Conv” layers with LeakyReLU and Sigmoid activations.

Suppose we have *C* input channel feature maps $\left[f_{1},f_{2},\;\ldots ,f_{C}\right]$, and then we squeeze them into a GAP layer to produce the channel-wise statistic $V\in R^{C\times 1\times 1}$. The cth element of V can be computed by the following expression:(11)$$V_{c}=\frac{1}{H\times W}\overset{H}{\underset{i=1}{\sum }}\overset{W}{\underset{j=1}{\sum }}f_{c}\left(i,j\right),$$
where $f_{c}\left(i,j\right)$ is the pixel value at the position (i,j) of the cth channel feature map $f_{c}$. To fully obtain the interdependence of each channel, we adopt a Sigmoid gating mechanism like [19] by two “1 × 1 Conv” layers forming a bottleneck with dimension-reduction and -increasing ratio r and use LeakyReLU as the activation function.
(12)$$f_{ca}=\delta \left(W_{U}*\sigma \left(W_{D}*V\right)\right),$$
where ∗ denotes the “1 × 1 Conv” operation, δ(⋅) and σ(⋅) represent the activation functions of Sigmoid and LeakyReLU, respectively. $W_{U}$ and $W_{D}$ represent the learned weights of the two “1 × 1 Conv” layers, respectively. Then, the final channel statistics maps can be obtained by
(13)$$F_{ca}=f_{conv}\times f_{ca},$$

In this way, our CA block can adaptively determine which feature channel should be focused or suppressed.

### 3.3. EdgeNet

As shown in Figure 4, inspired by the classic deep learning-based edge detection method [23], we proposed an edge enhanced network (EdgeNet) by modifying the edge detection module of RCF [18]. RCF contains 5 stages based on VGG16 [25], each of them receives a feature map through a series of stacked convolutional layers. To reduce the computational complexity of complex networks, the EdgeNet we proposed has only 3 stages, where “Dsconv” means depth-wise separable convolution and its operation is shown in the dotted box in Figure 4; “Deconv” means deconvolution, *k* × *k − n* (such as 3 × 3 − 64) means that the kernel size is *k*, and the number of filters is *n*.

Compared with [23], the adjustments we made are summarized: we deleted Stage 4, Stage 5, and the 2 × 2 pooling layer behind Stage 4 in RCF. Our EdgeNet consists of only 3 stages; we use depth-wise separable convolution layers to replace the ordinary 3 × 3 Conv layers in RCF to reduce the computational complexity of the network and achieve better learning of channel and region information.

### 3.4. Fusion

The structure of the Fusion part of our model is shown in Figure 5. This part first integrates $I_{\mathrm{sr}}$ and $I_{\mathrm{edge}}$ through a “*concat*” operation, and then performs a dimensionality reduction process on the fused image through a “1 × 1 Conv” operation to obtain the final reconstructed image. This method enables our network to fuse the enhanced edge information with the reconstructed image, so as to make full use of the edge information, thereby making the final reconstructed image edge clearer and sharper.

### 3.5. Loss Function

Most existing SISR methods use L1 or MSE loss, but both of these loss functions have certain shortcomings. This is also one of the main reasons that make it difficult to train some models and improve the reconstruction performance. The MSE loss will make the model very sensitive to outliers, which will easily lead to difficulty in model convergence during training. Although L1 loss is very robust to outliers, the gradient is also very large for small loss values. Therefore, we use charbonnier loss (a variant of L1 loss) proposed in LapSRN [36] to train our network instead of L1 or MSE loss, and we describe it in this section. Assume that $I_{\mathrm{LR}}$ and $I_{\mathrm{HR}}$ are the input LR image and the ground truth HR image, respectively; Θ is the network parameters. Let $I_{\mathrm{SR}}$ denote the final output of our network after using residual learning, and the loss function is defined by the following expression:(14)$$L_{s}\left(I_{\mathrm{HR}},I_{\mathrm{SR}};\Theta \right)=\frac{1}{N}\overset{N}{\underset{i=1}{\sum }}\rho \left(I_{\mathrm{HR}}^{i}-I_{\mathrm{SR}}^{i}\right),$$
where $\rho \left(x\right)=\sqrt{x^{2}+\epsilon^{2}}$, N is the number of training samples, and s is the upscale factor. We set ϵ as $1\times 10^{-3}$ based on experience. In the discussion section, we give an analysis of the results of training our model with different loss functions to further illustrate the effectiveness of the charbonnier loss we selected.

## 4. Discussion

**Loss function.** Here we use a set of comparative experiments to analyze the optimization performance of our model with three different loss functions. As shown in Figure 6, the convergence speed of the model optimized with charbonnier loss (green curve) is slightly faster than the MSE loss (orange curve), and finally the optimal PSNR value is obtained. Given comprehensive convergence speed and optimization performance, our research chooses charbonnier loss as the optimization function.

**Differences from SRFBN.** There is still a flexibility problem of comparing different deep learning networks. This section aims to explain the difference between the feedback mechanism in our EFANSR and SRFBN [14], as well as our rationality and superiority. There is only one feedback block in SRFBN and the feedback block is constructed by cascading several iterative up- and down-sampling convolutional layers, but our proposed network has *N* DSAN modules cascaded. Although SRFBN can use deep feature information to fine-tune the shallow input through the feedback mechanism, it cannot use multiple modules to cascade back deeper feature information to achieve fine-tuning of the shallow input like our network. In addition, we use local feature fusion technology to achieve full fusion utilization of the output features of each DSCA module.

## 5. Experimental Results

In this section, we first explain how to construct the datasets and training details. Next, we explore the effectiveness of our proposed attention mechanisms, feedback mechanisms, and edge enhancement network through ablation studies. Then, we compare our method with the most advanced current methods. Finally, the model parameters and performance of different methods are compared and analyzed.

### 5.1. Datasets and Metrics

We used DIV2K [37] as our training dataset. The DIV2K dataset contains 800 training images, 100 validation images, and 100 undisclosed test images. In this work, the LR images were, respectively, obtained by bicubic down-sampling the HR images with scaling factors 2×, 3×, and 4×. To fully use the data and improve the reconstruction performance, we augmented the training data by random horizontal flips and 90°, 180°, and 270° rotations like [12] did.

We used five benchmark datasets that are widely used as our test datasets: Set5 [38], Set14 [39], B100 [40], Urban100 [41], and Manga109 [42]. The Set5, Set14, and B100 datasets consist of natural scenes. The Urban100 set contains images from different frequency bands of urban scenes, and the Manga109 is a Japanese manga dataset.

To be consistent with the existing research work, we only calculated the peak signal-to-noise ratio (PSNR) and structural similarity (SSIM) on the luminance channel (Y channel in the YCbCr color space) of the SR image obtained by super-resolution reconstruction.

### 5.2. Training Details

We took the LR images in RGB format and the corresponding HR image as input and cropped the size of each input patch to 40 × 40. The networks were implemented on the PyTorch framework and trained on a NVIDIA 2070Ti GPU with a batch size of 16 and optimized using an Adam optimizer [43]. We set the parameters of the optimizer to: $\beta_{1}=0.9$, $\beta_{2}=0.99$, $\varepsilon =1\times 10^{-8}$. The learning rate was initialized to 1 × 10^−4^ and decreased by half when training reached 80, 120, and 160 epochs.

In the training process, the three parameters in Figure 2 were set as follows: the number of filters in all Conv layers was set to *n* = 64; the kernel size (*k*) and stride (*s*) change with the up-sampling scale factor, for 2×, *k* = 6, *s* = 2, for 3× and 4×, *k* = 3, *s* = 1. The parameter settings in EdgeNet are given in Figure 4. In the testing phase, to maximize the potential performance of the model, we adopted the self-ensemble strategy [44]. Based on experience and experiments, we found that such a parameter setting can achieve outstanding performance and balance training time and memory consumption.

### 5.3. Ablation Experiments

**Study of N.** Our DASRNet contains *N* DSCA blocks; in this sub-section, we study the effect of the value of *N* on reconstruction performance. We built models with different depths of *N* = 6, 8, and 10, and evaluated them quantitatively. The evaluation results are given in Table 1. It can be seen from the experimental results that the performance is relatively better when *N* = 8. Based on this, the *N* of all models in the subsequent experiments in this paper is eight.

**Results of attention mechanisms.** To visually illustrate the effectiveness of our proposed SA and CA blocks, we conducted an ablation experiment. After removing the corresponding modules from DSCA, the model was trained and tested on the DIV2K and Set5 datasets. Table 2 shows a quantitative evaluation of each module. As shown in Table 2, the baseline performance without SA and CA module is very poor. The best performance is when both SA and CA are introduced (PSNR = 34.11 dB).

Furthermore, we represent the model without SA and CA modules as Baseline and visualize the convergence of EFANSR and Baseline. As shown in Figure 7, the EFANSR model with attention mechanisms obtains lower training loss and better reconstruction performance. The results in Table 2 and Figure 7 both show that the attention mechanism we introduced can improve the reconstruction performance, and our model also has considerable generalization ability.

**Results of edge enhancement.** We demonstrate that our proposed EdgeNet can make the reconstructed image obtain sharper edges. We compared the network with EdgeNet module and the network without EdgeNet in this section and show the visualization results in Figure 8. The Network with EdgeNet module can generate more reasonable details of irregular areas and generate sharper edges.

We present the quantitative evaluation results on the Set5 dataset for 3× SR in Table 3. Through visual perception in Figure 8 and quantitative analysis in Table 3, it can be seen that the proposed EdgeNet module can obtain clear sharp edges of reconstructed images and improve the quality of reconstructed images.

### 5.4. Comparison with State-of-the-Art Methods

To verify the effectiveness of our proposed method, we conducted numerous comparative experiments on the benchmark datasets. We compared our network with the following classic methods: A+ [6], SRCNN [4], FSRCNN [7], VDSR [5], MemNet [8], EDSR [12], SREdgeNet [24], and SRFBN [14]. Our model is denoted as EFANSR (ours). We evaluated the SR results with PSNR, SSIM, and compared performance on 2×, 3×, and 4× SR. It is worth noting that our goal is to make the edge properties of SR images better while obtaining better quantitative evaluation indicators PSNR and SSIM.

We show the quantitative results in Table 4. Our method is slightly inferior to EDSR and SRFBN in PSNR/SSIM, but it is much better than other methods, and our model complexity is much lower than EDSR. In particular, compared with SREdgeNet [24], our performance is far superior to it, which also proves the effectiveness of our EdgeNet in improving the performance of refactoring.

We show the visual effect pictures on the Set5 and B100 datasets for 3× SR in Figure 9 and 4× SR on B100, Urban100, and Manga109 in Figure 10. Our method accurately recreates text information and parallel lines while retaining richer details. We observe that the reconstruction results of SRCNN [4] and VDSR [5] are very fuzzy and a lot of details are missing. The reconstruction results of SRFBN [14] still have artifacts caused by the mesh effect. Instead, our approach effectively preserves detailed information through attention mechanisms and edge enhancement, resulting in very sharp edges and better visual effects.

### 5.5. Model Parameters

We show the number of parameters versus the reconstruction performance of CNN-based methods in Figure 11. By the application of parameter sharing and deep-wise separable convolution, our EFANSR has 73% fewer parameters than MemNet [8], 79% fewer than SRFBN [14], and has only 2.4% of the EDSR [12]. Our proposed EFANSR achieves performance comparable to the state-of-the-art methods with lower complexity.

## 6. Conclusions

In this paper, we proposed an edge-enhanced image super-resolution network with a feedback mechanism and attention mechanism (EFANSR), which is composed of DASRNet, EdgeNet, and the final fusion part. Our DSCA module fully uses high-frequency detail information by combining spatial attention (SA) and channel attention (CA). The introduced feedback mechanism enables our network to adjust the input by effectively feeding back the high-level output information to the low-level input. The edge enhancement network (EdgeNet) realizes the extraction and enhancement of different levels of edge information through several convolution channels with different receptive fields. Through the analysis of model complexity (such as model parameters) and numerous comparative experiments, it is fully proven that our method achieves performance comparable to the state-of-the-art methods with lower complexity, and the reconstructed image edges are sharper and clearer.

## Figures and Tables

**Figure 1 sensors-21-02064-f001:**
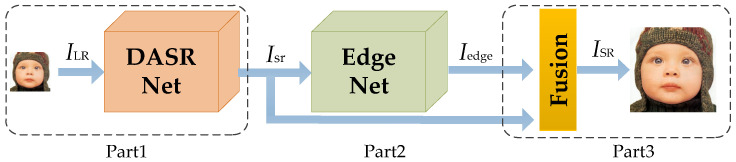
Our proposed enhanced feedback attention image super-resolution network (EFANSR) framework comprises three parts: a dense attention super-resolution network (DASRNet), EdgeNet, and Fusion.

**Figure 2 sensors-21-02064-f002:**
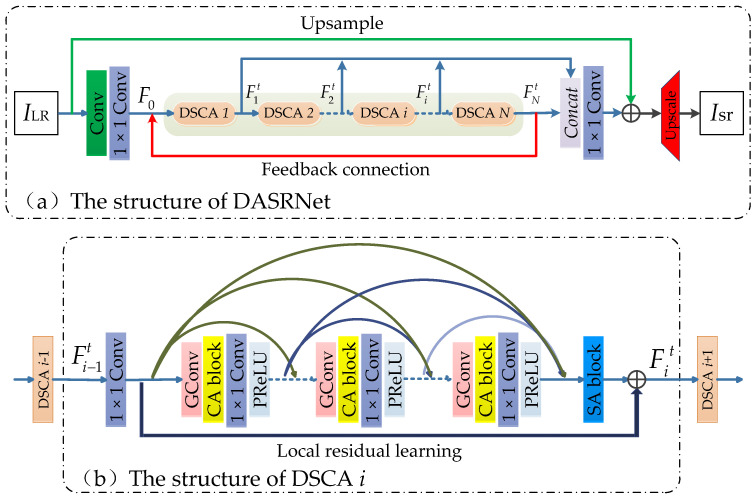
Our proposed DASRNet. The structure of DSCANet is shown in (**a**) and the structure of the dense spatial and channel attention module (DSCA) module is shown in (**b**). ⨁ means element-wise summation; the red and green arrows in (**a**) indicate feedback connections and global residual skip connections, respectively; “GConv” in (**b**) is composed of “deconvolution + PReLU + Conv + PReLU”.

**Figure 3 sensors-21-02064-f003:**
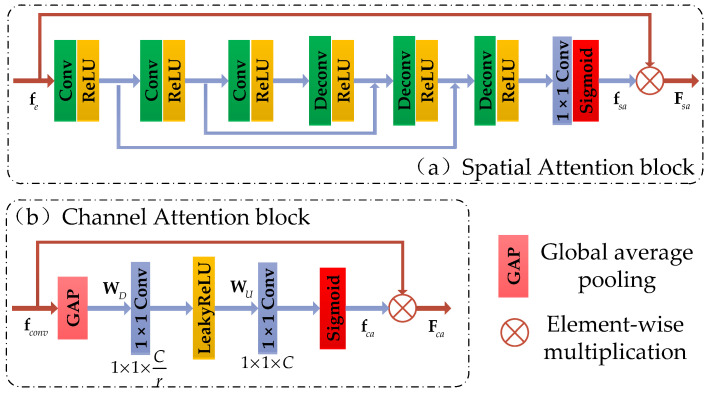
The space attention block and channel attention block in the DSCA module.

**Figure 4 sensors-21-02064-f004:**
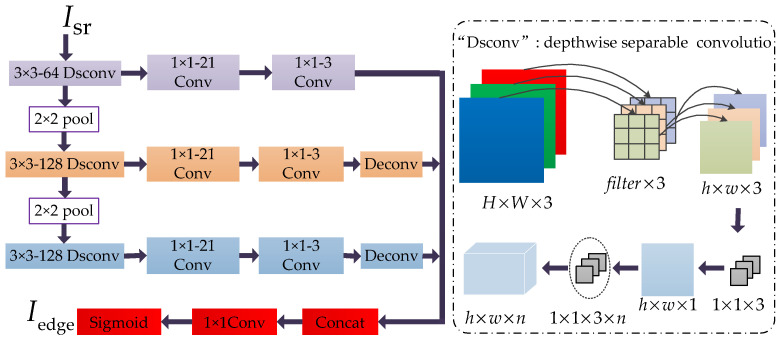
Our proposed edge enhanced network (EdgeNet). “Dsconv” means depth-wise separable convolution, and its calculation flow is shown in the inside of the dotted box. “Deconv” means deconvolution, and “pool” means pooling layer. “*k* × *k* − *n*” means the kernel size of this layer is *k* and the number of filters is *n*.

**Figure 5 sensors-21-02064-f005:**
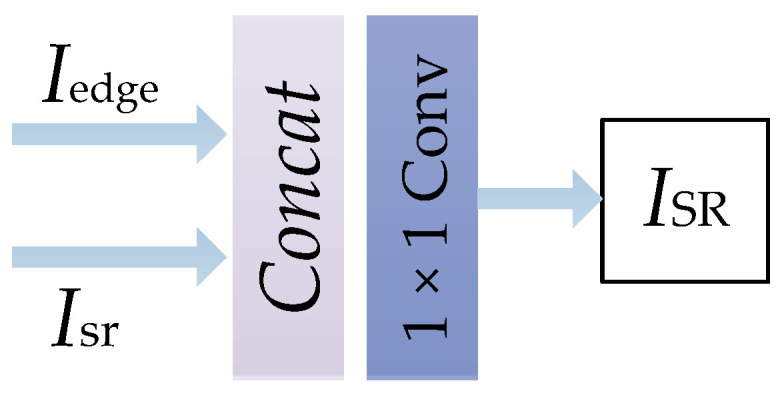
The structure of the Fusion part.

**Figure 6 sensors-21-02064-f006:**
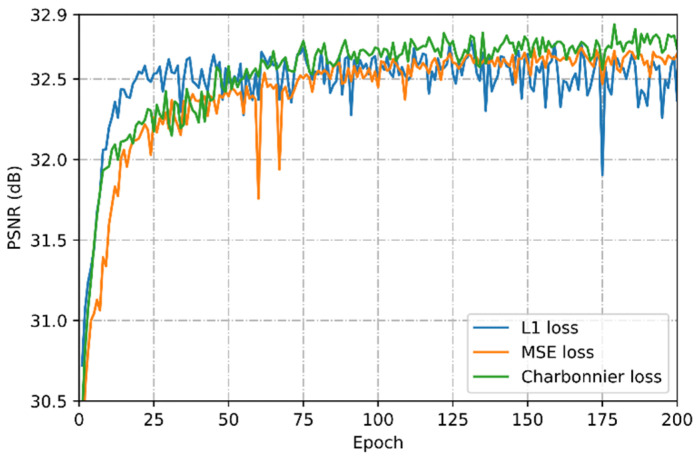
Performance analysis of different loss functions on the DIV2K dataset with 4×.

**Figure 7 sensors-21-02064-f007:**
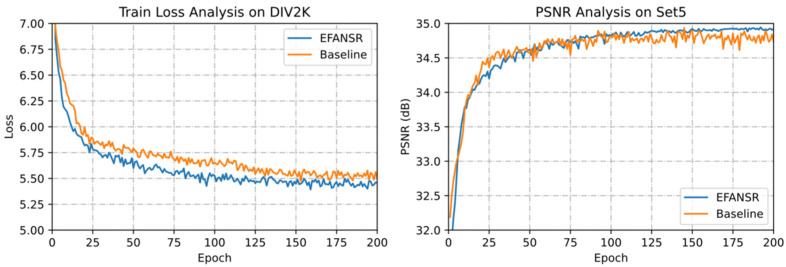
Convergence analysis of EFANSR (blue curves) and Baseline (orange curves) with 3×. On the left is the training loss curve on the DIV2K dataset, and on the right is the peak signal-to-noise ratio (PSNR) curve on the Set5 dataset.

**Figure 8 sensors-21-02064-f008:**
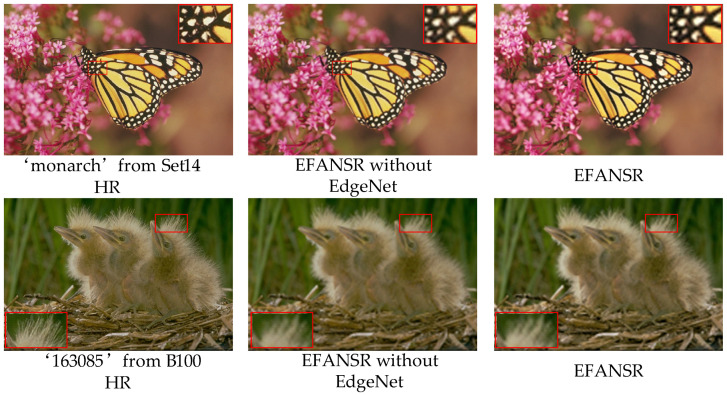
Visual comparison for EdgeNet. We compared the results of models with and without EdgeNet on 3× SR.

**Figure 9 sensors-21-02064-f009:**
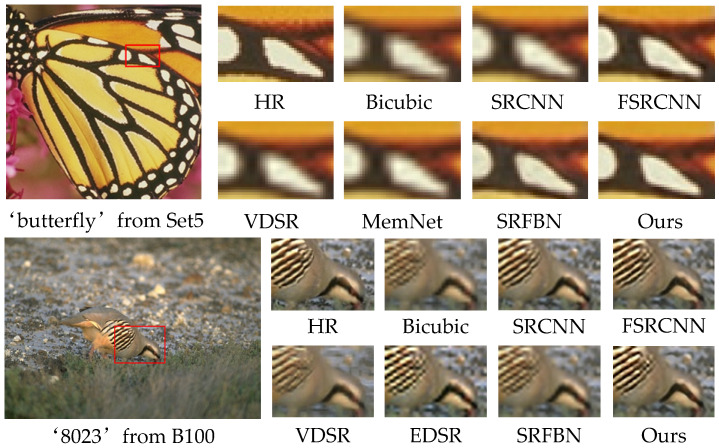
Visual comparison on the Set5 and B100 datasets for 3× SR.

**Figure 10 sensors-21-02064-f010:**
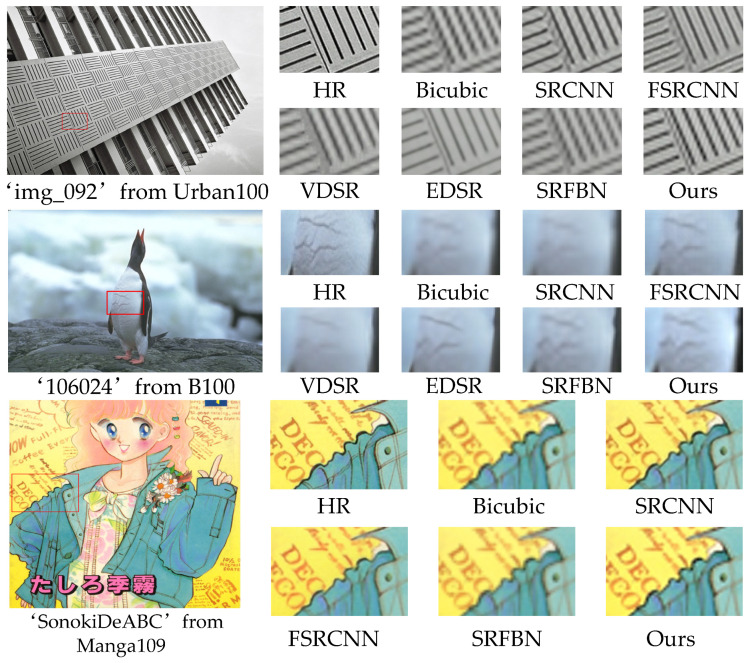
Visual comparison on the Urban100, B100, and Manga109 datasets for 4× SR.

**Figure 11 sensors-21-02064-f011:**
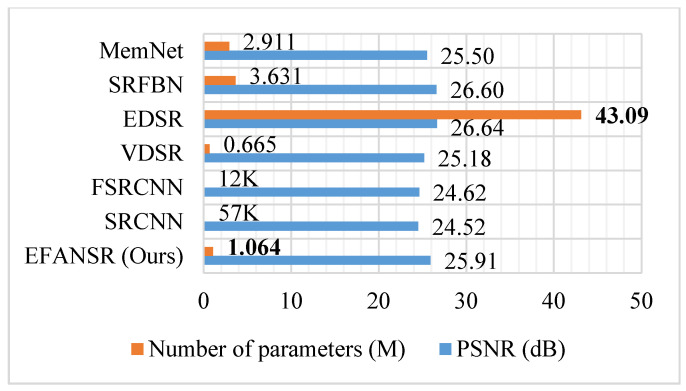
Comparison of the number of network parameters and the performance on the Urban100 dataset for 4× SR.

**Table 1 sensors-21-02064-t001:** Quantitative results of the number of DSCA blocks *N*. We compare three different models on the Set14 and B100 dataset for 4× SR.

Model	Parameters	Set14 PSNR/SSIM	B100 PSNR/SSIM
*N* = 6	1.056 M	28.33/0.7867	27.42/0.7409
*N* = 8	1.064 M	28.40/0.7869	27.55/0.7410
*N* = 10	1.072 M	28.31/0.7797	27.57/0.7412

**Table 2 sensors-21-02064-t002:** Quantitative results of attention mechanisms. We compared four different models (whether they contain spatial attention or channel attention) on the Set5 dataset for 3× SR. The model that includes both channel and spatial attention achieves superior performance.

Scale	Spatial Attention (SA)	Channel Attention (CA)	PSNR	SSIM
3×	×	×	33.84	0.9248
	√	×	33.67	0.9239
	×	√	33.99	0.9250
	√	√	34.11	0.9277

**Table 3 sensors-21-02064-t003:** Quantitative results of edge enhancement. We compared two different models (whether they contain EdgeNet) on the Set5 dataset for 3× SR.

Scale	EdgeNet	PSNR	SSIM
3×	×	32.99	0.9125
	√	34.11	0.9277

**Table 4 sensors-21-02064-t004:** Quantitative results of state-of-the-art single image super-resolution (SISR) methods. The average results of PSNR/structural similarity (SSIM) with 2×, 3×, and 4× on datasets Set5, Set14, B100, Urban100, and Manga109.

Dataset	Scale	Bicubic	A+	SRCNN	FSRCNN	VDSR	MemNet	EDSR	SREdgeNet	SRFBN	ours
Set5	2×	33.66/0.9299	36.54/0.9544	36.66/0.9542	37.00/0.9560	37.53/0.9587	37.78/0.9597	38.11/0.9606	-/-	38.11/0.9609	37.81/0.9509
	3×	30.39/0.8682	32.58/0.9088	32.75/0.9090	33.16/0.9139	33.66/0.9213	34.09/0.9248	34.65/0.9282	-/-	34.70/0.9292	34.11/0.9277
	4×	28.42/0.8104	30.30/0.8590	30.48/0.8628	30.71/0.8660	31.35/0.8838	31.74/0.8893	32.46/0.8968	31.02/0.8920	32.47/0.8983	31.74/0.8892
Set14	2×	30.24/0.8688	32.28/0.9056	32.45/0.9067	32.63/0.9089	33.03/0.9124	33.28/0.9142	33.92/0.9195	-/-	33.82/0.9196	33.56/0.9202
	3×	27.55/0.7742	29.13/0.8188	29.30/0.8215	29.43/0.8242	29.77/0.8314	30.00/0.8350	30.52/0.8462	-/-	30.51/0.8461	29.99/0.8469
	4×	26.00/0.7027	27.43/0.7520	27.50/0.7513	27.59/0.7549	28.02/0.7676	28.26/0.7723	28.80/0.7876	27.24/0.7800	28.81/0.7868	28.40/0.7869
B100	2×	29.56/0.8431	31.21/0.8863	31.36/0.8879	31.51/0.8920	31.90/0.8960	32.08/0.8978	32.32/0.9013	-/-	32.29/0.9100	31.89/0.9004
	3×	27.21/0.7385	28.29/0.7835	28.41/0.7863	28.53/0.7910	28.83/0.7980	28.96/0.8001	29.25/0.8093	-/-	29.24/0.8084	28.90/0.8089
	4×	25.96/0.6675	26.82/0.7100	26.90/0.7101	26.97/0.7150	27.29/0.7260	27.40/0.7281	27.71/0.7420	27.06/0.7380	27.72/0.7409	27.55/0.7410
Urban100	2×	26.88/0.8403	29.20/0.8938	29.50/0.8946	29.87/0.9020	30.77/0.9140	31.31/0.9195	32.93/0.9351	-/-	32.62/0.9328	31.73/0.9326
	3×	24.46/0.7349	26.03/0.7973	26.24/0.7989	26.43/0.8080	27.14/0.8280	27.56/0.8376	28.80/0.8653	-/-	28.73/0.8641	28.07/0.8546
	4×	23.14/0.6577	24.34/0.7200	24.52/0.7221	24.62/0.7280	25.18/0.7524	25.50/0.7630	26.64/0.8033	25.82/0.7910	26.60/0.8015	26.05/0.8029
Manga109	2×	30.80/0.9339	-/-	35.60/0.9663	36.65/0.9709	37.22/0.9750	37.72/0.9740	39.10/0.9773	-/-	39.08/0.9779	38.09/0.9778
	3×	26.95/0.8556	-/-	30.48/0.9117	31.10/0.9210	32.01/0.9340	32.51/0.9369	34.17/0.9476	-/-	34.18/0.9481	33.69/0.9461
	4×	24.89/0.7866	-/-	27.58/0.8555	27.90/0.8610	28.83/0.8870	29.42/0.8942	31.02/0.9148	-/-	31.15/0.9160	30.50/0.9150

## Data Availability

Not applicable.

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
