# Peer review of "Edge-Enhanced with Feedback Attention Network for Image Super-Resolution"

_sensors, 2021, doi:10.3390/s21062064_

Round 1
Reviewer 1 Report
The paper presents a new method for super-resolution reconstruction. It comprises 3 stages, a dense attention super-resolution network (DASRNet), an edge detection and enhancement network (EdgeNet), and a fusion reconstruction module.
The three stages are inspired in previous works, which are cited along the paper.
The results (visual, PSNR, SSIM) are comparable to other pervious works, with the benefit of use less parameters, which I may assume reduce the training and overall processing times.
The experiments are similar to other works, they allows a fair comparison.
Scientific content
The proposed method includes 3 stages, but the final one is not described in the paper. How is the Fusion made? in line 210, the authors talk about a fusion, but I seems that it refers to the last steps of DASRNet
In line 204, the authors explain the DSCA module. They explain a recursive part, that is not clear in figue 2. Moreover, the iterations are referenced as t, while the figure use T. The unfolding is not needed.
In line 233, even though it is grammarly correct, I recommend change the writing from "gradient vanishment" to "vanishing gradient" to be consistent with literature about this problem.
In line 266, it should be cleared that "the dotted box" refers to figure 4.
In the experiments, the claims are based on numerical results of PSNR and SSIM, however, many of the differences are not significant. This is, the proposed method is just slightly better/worse than best previous works. However, this results are achieved with significantly less (network) parameters. Hence, the emphasis should be about the network complexity and not on the PSNR/SSIM.
The references should be reviewed, this is because, in line 299, reference [37] do not correspond to the DIV2K dataset https://data.vision.ee.ethz.ch/cvl/DIV2K/
@InProceedings{Agustsson_2017_CVPR_Workshops,
author = {Agustsson, Eirikur and Timofte, Radu},
title = {NTIRE 2017 Challenge on Single Image Super-Resolution: Dataset and Study},
booktitle = {The IEEE Conference on Computer Vision and Pattern Recognition (CVPR) Workshops},
month = {July},
year = {2017}
}
Language and Style
The paper needs a language and style review, there are some mistakes, like
* the authors use many names (using initial letters) which are not expanded, at least the first time, like FSRCNN, VDSR, EDSR, RDN, etc.
* ....The final fart is to merge .... in line 21
* .....SRGAN [21] combining perceptual loss [37] 76 and adversarial loss [22] is proposed. in line 76
*.....but there is no SR method to solve 155 this problem using edge detection until Kim et al. proposed SREdgeNet [24]. in line 155
and others.
The style should be reviewed, for example, Caption for Table 3 looks like a text paragraph, and it should only describe the table contents: "Table 3. Quantitative results of edge enhancement for two different models (whether they contain EdgeNet) on the Set5 dataset for 3× SR." A more complete explanation and performance claims should be part of the document's body.
Reviewer 2 Report
Overall, I am happy with the main aspects of this submission, in particular the novelty and soundness of the approach, and its evaluation. These are the most important things, so I am happy to recommend that this work should be considered further for acceptance and publication.
Having said the above, there are a number of issues in the writing and presentation which really ought to be corrected or improved before final acceptance is made. Here are some examples of the kinds of issues that I noticed:
- In the abstract, it is not clear what "the attention mechanism and feedback mechanism" is referring to. The sentence and the concepts within it are introduced before any prior context which makes it difficult to understand when first read (they do make sense retrospectively).
- Still in the abstract, in "Besides, the existing methods still have serious artifacts and edge blurring" it is not methods that have artefacts and blur but the output thereof; also "besides" should be replaced with "additionally" (or "in addition").
- The claim that "Single image super-resolution (SISR)" is a "classic low-level computer" is questionable. How can it be low level if it is based on the learning of higher level semantic content? In short, SISR is not always or inherently low level. This sentence should be rephrased at the very least.
- In "we carefully designed", "carefully" should be removed.
- The claim that "image edge detection has been in-depth research in recent decades" is at least questionable (and likely just wrong) and should be rephrased or removed (there are also grammatical errors in the sentence).
- The authors should not write "such as [33]." - it reads the same as "such as." as citations are not part of main text. Rather, this should be written as "as done by Jones [33]".
- How can "accuracy of these methods in practical applications is often low"? Edges are by definition appearance based entities, not something that has intrinsic ground truth in the real world. I suspect that the authors are not expressing their thoughts well here.
- What do the authors mean by "predict edge strength"? Edge strength just is, it is not something that needs prediction or that can be predicted.
- Explanations of various choices (e.g. "We use charbonnier loss") should be made; there are many examples like this when some decision is merely stated without any justification.
Round 2
Reviewer 2 Report
Save for a number of linguistic mistakes which remain and which can be handled by the production staff, the authors have addressed my concerns adequately.